# Mitochondrial Consequences of Organ Preservation Techniques during Liver Transplantation

**DOI:** 10.3390/ijms22062816

**Published:** 2021-03-10

**Authors:** Tamara Horváth, Dávid Kurszán Jász, Bálint Baráth, Marietta Zita Poles, Mihály Boros, Petra Hartmann

**Affiliations:** Institute of Surgical Research, University of Szeged, 6724 Szeged, Hungary; horvath.tamara@med.u-szeged.hu (T.H.); jasz.david.kurszan@med.u-szeged.hu (D.K.J.); barath.balint@med.u-szeged.hu (B.B.); poles.marietta.zita@med.u-szeged.hu (M.Z.P.); boros.mihaly@med.u-szeged.hu (M.B.)

**Keywords:** liver transplantation, graft preservation, mitochondrial functions, animal studies, meta-analysis

## Abstract

Allograft ischemia during liver transplantation (LT) adversely affects the function of mitochondria, resulting in impairment of oxidative phosphorylation and compromised post-transplant recovery of the affected organ. Several preservation methods have been developed to improve donor organ quality; however, their effects on mitochondrial functions have not yet been compared. This study aimed to summarize the available data on mitochondrial effects of graft preservation methods in preclinical models of LT. Furthermore, a network meta-analysis was conducted to determine if any of these treatments provide a superior benefit, suggesting that they might be used on humans. A systematic search was conducted using electronic databases (EMBASE, MEDLINE (via PubMed), the Cochrane Central Register of Controlled Trials (CENTRAL) and Web of Science) for controlled animal studies using preservation methods for LT. The ATP content of the graft was the primary outcome, as this is an indicator overall mitochondrial function. Secondary outcomes were the respiratory activity of mitochondrial complexes, cytochrome c and aspartate aminotransferase (ALT) release. Both a random-effects model and the SYRCLE risk of bias analysis for animal studies were used. After a comprehensive search of the databases, 25 studies were enrolled in the analysis. Treatments that had the most significant protective effect on ATP content included hypothermic and subnormothermic machine perfusion (HMP and SNMP) (MD = −1.0, 95% CI: (−2.3, 0.3) and MD = −1.1, 95% CI: (−3.2, 1.02)), while the effects of warm ischemia (WI) without cold storage (WI) and normothermic machine perfusion (NMP) were less pronounced (MD = −1.8, 95% CI: (−2.9, −0.7) and MD = −2.1 MD; CI: (−4.6; 0.4)). The subgroup of static cold storage (SCS) with shorter preservation time (< 12 h) yielded better results than SCS ≥ 12 h, NMP and WI, in terms of ATP preservation and the respiratory capacity of complexes. HMP and SNMP stand out in terms of mitochondrial protection when compared to other treatments for LT in animals. The shorter storage time at lower temperatures, together with the dynamic preservation, provided superior protection for the grafts in terms of mitochondrial function. Additional clinical studies on human patients including marginal donors and longer ischemia times are needed to confirm any superiority of preservation methods with respect to mitochondrial function.

## 1. Introduction

### 1.1. Background

Liver transplantation (LT) is the treatment of choice for patients with end-stage liver disease. From the first human LT performed by Thomas Starzl in 1963, advances in surgical technology and effective immunosuppressive agents have increased the five-year survival of transplanted patients by over 75% [1]. The success of LT, however, is limited by a shortage of donor organs compared to waiting list demand. Efforts to expand the donor pool have included the use of suboptimal, so-called extended criteria donor (ECD) grafts, which were previously considered unsuitable for transplantation, allowing the use of organs after prolonged cold ischemia times (CIT), inclusion of older donors, donation after cardiac death (DCD) or hepatic steatosis [2] (Table 1).

The subsequent increase in the incidence of primary graft non-function (PNF), early allograft dysfunction (EAD) and biliary complications all required the modification of technical protocols [3]. Alternative preservation methods attempt to reduce graft damage via two main approaches: (1) maintaining organ perfusion after graft procurement using pulsatile machine perfusion (MP) with fluids and/or gas, and (2) modifying the temperature of the perfusate.

### 1.2. Organ Preservation Techniques

The principle of simple/static cold storage (SCS) was first introduced for kidney grafts by Geoffrey Collins in 1969 and became the gold standard storage option for organ transplantation [4]. SCS remained the clinical standard of care for liver graft preservation, using modern preservation solutions, including the University of Wisconsin (UW), and the Histidine-Tryptophan-Ketoglutarate (HTK) and Celsior solutions on ice [5]. The purpose of SCS is to lower metabolic activity and oxygen demand before transplantation, but tissue injury is still present, especially with ECD grafts or prolonged CIT (Figure 1).

At present the accepted CIT for clinical LT is 12 h; however, CIT ≥ 4 h is associated with considerably lower graft survival than CIT < 4 h [6]. Negative outcomes include increased risk of PNF, graft failure and patient death, along with reduced long-term graft survival. Hypothermic machine perfusion (HMP), where the donor organ is continuously perfused with preservation solution, was introduced in an attempt to limit these undesirable consequences [7]. While the increased distribution of nutrients and the clearance of toxic metabolites with HMP provide certain benefits, the results of studies comparing HMP and SCS are controversial [8]. Additional shortcomings of HMP, such as higher expenses and complexity make this approach less feasible than SCS. As a further step to preventing the negative effects of cold-induced injuries, organ preservation at a higher preservation temperature—closer to that of physiological conditions—was developed. The current technique is perfusion of the graft with oxygenated autologous blood, erythrocyte-based solutions, or a cellular solution at normothermic (35–38 °C) (NMP) or subnormothermic (25–34 °C) (SMP) temperatures [8]. Several studies have shown that NMP improves outcomes in terms of lower early allograft dysfunction (EAD) and aspartate transaminase (AST) levels as compared to SCS [9,10,11,12]. However, problems with NMP are numerous, including technical complexity, and economic and logistical problems, as no standard perfusion apparatus or protocol exists as of today. Due to these challenges, development of simpler alternative techniques is of great clinical interest.

### 1.3. Mechanism of IR-Induced Mitochondrial Dysfunction

Protection of mitochondrial functions should be considered a main strategy for graft preservation. Mitochondria are sites of high-energy phosphate synthesis and calcium stores, and activate signaling pathways that impact cell fate directly. We, and others, have shown that hepatic IR induces damage to mitochondrial structure and function, which contributes to poor outcomes upon transplantation [13,14,15]. The mechanism of mitochondrial metabolic changes upon ischemia is relatively well-known, the interruption of blood flow and the concomitant hypoxia reduces mitochondrial electron transport chain (ETC) activity, and cells switch to anaerobic glycolysis. In the absence of oxygen, the highly reduced CoQ pool passes electrons onto fumarate by reversal of succinate dehydrogenase (respiratory complex II), leading to succinate accumulation; a metabolic marker of ischemia [16,17,18] (Figure 2).

The lack of ATP production by oxidative phosphorylation also inhibits the activity of ATP-dependent membrane Na^+^/K^+^ ATPases. The increase in extramitochondrial Na^+^ impairs other transporters, including Na^+^/Ca^2+^ efflux through the plasma as well as mitochondrial membranes. The accumulation of Ca^2+^ in the cytoplasm, and eventually in the mitochondrial matrix, has been shown to increase mitochondrial permeability, contributing to the production of reactive oxygen species (ROS), both of which are exacerbated during reperfusion [17]. Upon reperfusion, the leaked electrons reduce the newly present oxygen due to impaired ETC activity, thus leading to an excess of ROS that cannot be eliminated. Therefore, the ROS-mediated damage to ETC complexes leads to more pronounced ATP depletion, which can ultimately cause cell death [18]. ROS also damage mitochondrial membrane lipids, causing the opening of mitochondrial permeability transition pores (MPTPs) along with Ca^2+^-induced mitochondrial swelling. Due to increased mitochondrial membrane permeability, mediators of the intrinsic apoptotic pathway are released into the cytoplasm, thus, initiating apoptosis.

### 1.4. The Impact of Mitochondrial Damage on Transplantation

The role of IR-induced mitochondrial damage as one of the key contributors in allograft functions is increasingly recognized [19,20,21]. Our research group previously demonstrated that SCS of heart grafts resulted in decreased mitochondrial oxidative phosphorylation capacity and cytochrome c release, together with increased transcription of proapoptotic proteins [14]. These changes were associated with a parallel decrease in myocardial contractility. Mitochondrial gene expression changes in cardiac and renal transplantation recipients further support these findings in association with allograft injury and rejection [22,23]. Disturbances of mitochondrial oxidative pathways are demonstrated during the acute rejection process, and there is good evidence for decreased glycolytic enzyme activity in the graft [24]. When the post-transplant mitochondrial function in the renal and heart biopsies were evaluated by high-resolution respirometry, the results showed declined oxidative capacity [14,25,26]. The role of cold storage in mitochondrial damage has also been demonstrated in SCS renal grafts; without transplantation, reduced complex I, II and III activities were demonstrated along with decreased expression of the proteins controlling mitochondrial fusion and fission [26,27].

Unlike cardiac or renal transplantation, human studies on mitochondrial function in LT are very scarce. In aerobic conditions, the liver derives its energy primarily from the mitochondria, which constitute 20–25% of the total hepatic cell volume with 1000 to 2000 mitochondria per cell; optimizing hepatic metabolic function before LT is, therefore, an important task because this factor is closely linked to post-transplant graft survival and positive outcomes [28]. Nevertheless, no systematic analysis of mitochondrial function in humans exists. So far, only animal models of LT have provided data on mitochondrial metabolic changes, respiratory activity of mitochondrial complexes or the release of pro-apoptotic proteins. Furthermore, there is no clear-cut evidence for optimal preservation methods, and a comprehensive quantitative analysis is also missing in this respect.

### 1.5. Aims

The purpose of this review is to explore the impact of different liver preservation techniques on mitochondrial functional changes, which may contribute to the outcome of LT. We will discuss the role of temperature during the storage of the graft and the most frequent clinical applications of machine perfusion (MP) in animal experiments. Animal models of LT play a crucial role in all stages of developing future clinical strategies and therapeutic interventions for human health and diseases, since a wide range of experimental data can be safely obtained from them. Therefore, we will provide a summary of current relevant animal studies in the field, with a special focus on mitochondrial functional parameters. The primary outcome of our study is to evaluate the ATP content of liver allografts, by which the efficiency of oxidative phosphorylation can be evaluated, and this includes overall mitochondrial bioenergetics as well. Other mitochondrial functional parameters, such as mitochondrial respiratory complex (I–V) activities, cytochrome c and AST release and mitochondrial apoptosis markers, will also be evaluated as further outcomes.

## 2. Results

### 2.1. Eligible Studies and Study Characteristics

A total of 4598 records were identified through our search strategy in January 2021. After excluding duplicates, the remaining 2508 articles were screened by title. A total of 201 abstracts were assessed, and 150 publications were enrolled in the final, comprehensive full-text analysis. After this, 25 records ultimately met our eligibility criteria. Table 2 shows a summary of the included studies. The flowchart of study enrolment is shown in Figure 3. 

In order to reduce any bias originating from the study protocols of the included studies, we excluded machine perfusion (MP) groups using non-oxygenated perfusion solutions or protocols where the length of perfusion was shorter than one hour. We only included MP groups where the grafts were perfused for the entire duration of preservation or where MP was preceded by static cold storage (SCS). We excluded warm ischemia (WI) groups with a warm ischemic time longer than 90 min. We also excluded all groups (1) which compared a modified version of a preservation solution to the standard solution, (2) which were preconditioned with ischemia and (3) which received some form of treatment prior to or during preservation.

### 2.2. Study Bias

We used the Cochrane risk-of-bias analysis to assess the risk of bias for each study (Figure 4).

### 2.3. Findings of Meta-Analysis

A meta-regression analysis of tissue ATP content (Figure 5) showed a significant difference (*p* < 0.05) in the SCS < 12 h (MD = −1.51; 95% CI = −2.92 to −0.09), SCS ≥ 12 h (MD = −3.09; 95% CI = −4.13 to −2.06) and WI (MD = −1.77; 95% CI = −2.88 to −0.65) groups compared to the control group. There was, however, no significant difference in the HMP (*p* = 0.13; MD = −1.00; 95% CI = −2.3 to 0.3), NMP (*p* = 0.097; MD = −2.09; 95% CI = −4.57 to 0.38) and subnormothermic machine perfusion (SNMP) (*p* = 0.308; MD = −1.10; 95% CI = −3.22 to 1.02) groups. 

An identical analysis of CI-linked mitochondrial respiratory activity revealed a significant decrease (*p* < 0.05) in respiratory control ratio (RCR) (MD = −4.24; 95% CI = −7.79 to −0.69) and OxPhos activity (MD = −13.10; 95% CI = −15.22 to −10.98) in the WI group, but not in Leak state (*p* = 0.823; MD = −0.1; 95% CI = −0.98 to 0.78) (Figure 6). In the SCS ≥ 12 h group, there was a significant increase (*p* < 0.05) in Leak state (MD = −1.00; 95% CI = 0.62 to 1.38), while RCR (*p* = 0.492; MD = −0.89; 95% CI = −3.44 to 1.65) and OxPhos activity (*p* = 0.089; MD = −0.1; 95% CI = −0.98 to 0.78) were not significantly different compared to the control group. There was no significant difference in RCR (*p* = 0.927; MD = −0.16; 95% CI = −3.28 to 3.6), OxPhos activity (*p* = 0.443; MD = 0.41; 95% CI = −0.64 to 1.46) or Leak state (*p* = 0.566; MD = 0.7; 95% CI = −0.17 to 0.31) in the SCS < 12 h group.

Analysis of cellular damage markers revealed a significant increase (*p* < 0.05) in AST levels in the WI group (MD = 2943.7; 95% CI = 831.55 to 5055.85), but no significant difference was observed in SCS ≥ 12 h (*p* = 0.793; MD = 480.0; 95% CI = −3108.8 to 4068.8) and HMP (*p* = 0.869; MD = 426.5; 95% CI = −4643.43 to 5496.43) groups (Figure 7). Cytochrome c content showed no significant difference in SCS < 12 h (*p* = 0.736; MD = −0.01; 95% CI = −0.09 to 0.06) or SCS ≥ 12 h (*p* = 0.564; MD = −0.02; 95% CI = −0.07 to 0.04) groups compared to the control group.

## 3. Discussion

This study reviews the latest preclinical evidence on ischemia/reperfusion (I/R) injury in LT—which can disrupt the normal activity of mitochondria in the hepatic parenchyma—and graft preservation methods, which have an impact on I/R injury. A network meta-analysis comparison provided good evidence that the present gold standard preservation, SCS, is outperformed by SNHMP and HMP in providing good functional outcomes with regard to ATP content. Additionally, WI alone, without cold storage, displayed worse results than SCS, suggesting the protective role of cooling in mitochondrial metabolism. According to the results, prolonged cold ischemia in SCS ≥ 12 h deteriorated mitochondrial leak respiration and ATP content of the grafts. Lower AST levels in HMP revealed that preservation of mitochondrial respiration enhances functional recovery and decreases cellular necrosis of the graft.

### 3.1. Principal Findings and Comparison with Other Studies

Until now, no systematic review has addressed the role of mitochondrial function in LT. All the published studies that have compared the preservation techniques of liver grafts mainly focused on the feasibility and safety of the technology investigating clinical outcomes such as EAD, PNF and ischemic-type biliary lesion (ITBL) [52,53,54,55,56,57,58].

The first clinical study that used HMP prior to transplantation was carried out by Guarrera et al. in a non-actively oxygenated model of HMP [52]. Since then, the concept of hypothermic oxygenated machine perfusion (HOPE), with active oxygenation of the perfusate and dual hypothermic oxygenated machine perfusion (D-HOPE) was developed, with beneficial effects on post-transplant biliary complications, particularly ITBL [53,54]. Subsequently, an increasing number of comparative studies on MP and cold storage (CS) have been reported; however, the benefits on outcomes were inconclusive without pooled analyses. Animal studies are more comparable, and two meta-analyses of animal studies concluded that MP preservation is superior to SCS in terms of reducing hepatocellular injury and biliary injury [55,56]. The largest clinical trial involving NMP as a preservation strategy was recently published by Nasralla et al. It found that the 30-day graft survival of NMP grafts was similar to that of SCS grafts and the AST level within the first seven post-operative days was lower [57]. A recent study comparing HMP and NMP with SCS, revealed that HMP but not NMP produced significant protective effects on EAD and biliary complications as compared to SCS [58].

Several studies found that the ATP content of the graft is predictive of post-transplant outcomes, so we chose this as the primary outcome of our study [16,59,60,61,62,63]. This meta-analysis demonstrated the superiority of HMP and SNMP over SCS in preserving ATP levels. This finding is consistent with previous studies, where ATP was also a marker of the viability of MP-treated grafts [53,54,62,63,64]. Another result of this meta-analysis is that SCS < 12 h demonstrated better results in terms of ATP preservation than WI grafts. Several studies showed differences in the cellular and molecular mechanisms between cold and WI [65,66]. At the level of the graft, hepatocytes were mainly affected during WI, while hepatic sinusoidal endothelial cells were more susceptible to cold ischemia [67]. The background of the beneficial metabolic effects of cold ischemia is that it (i) preserves the ATP/ADP ratio and adenine nucleotides better, (ii) inhibits a high NADH/NAD^+^ ratio and (iii) avoids excessive succinate accumulation [20] (Figure 8). 

The decisive role of succinate in warm I/R injury was demonstrated in an animal model, where succinate alone was sufficient to cause extensive damage to the graft, independently of other effects of WI [21].

Our meta-analysis found significant differences when comparing mitochondrial respiration under cold and warm conditions of graft preservation. The maximal ADP-stimulated O_2_ consumption rate (OxPhos, state 3) in the graft, which indicates the capacity to produce ATP through the oxidative phosphorylation pathway, was severely damaged in WI as compared to SCS. The RCR, which detects any changes in oxidative phosphorylation capacity related to tightness of mitochondrial coupling, was also affected. This is consistent with findings from previous reviews and studies [64]. Such remarkable differences in mitochondrial electron transport are related to a unique mitochondrial response under cold conditions. Hypothermia slows mitochondrial activity, which evolves with decreased proton motive force or with a delayed transition of the de-activated (D) from complex I to its active form [67,68] (Figure 9).

### 3.2. Strengths and Limitations

We made an effort to provide objective results. First, the study searched four main databases, and two investigators examined the experiments to make certain that all of the relevant studies were included in the research. Secondly, the SYRCLE Risk of Bias analysis was used independently by two investigators to reduce bias in assessing the methodological quality of the studies, and they ultimately summarized the key results. This method has been adjusted for particular aspects of bias that play a role in animal intervention studies and a low bias of the included studies was revealed. Thirdly, two investigators implemented the extracted data to ensure that all of the outcomes were accurately extracted and synthesized from the reported experiments.

Our study has limitations, some of them originate in the method that we used for the analysis. We employed a network meta-analysis, which is widely used for aggregating results of clinical trials to make direct and indirect inferences about treatment effects. In contrast to traditional meta-analyses, which aggregate studies on the same study question, network meta-analyses also involve studies on different study questions which are linked by pairwise same-treatment groups [69]. Others share the limitations of the original studies. This study aimed to investigate the mitochondrial effects of different graft preservation techniques in liver transplantation; however, other mechanisms might exist during the complex process of the graft recovery. Other predictors of poor outcome, such as surgery time, differences in the operative procedure, the effects of different preservation solutions and application of scavengers, were not considered in this meta-analysis [70,71]. Storage time is often cited in the literature as a strong predictor, but we were only able to compare the effect of shorter (<12 h) and longer (≥12 h) storage times within the SCS group. Furthermore, none of the studies mentioned the sample size calculation process, which is necessary to achieve scientific objectives without missing biologically important effects.

### 3.3. Implications for Practice and Conclusion

To date, human clinical results have strengthened the association between disturbances in mitochondrial bioenergetics and the acute rejection process, thus, raising the possibility of establishing a rejection score containing data predictive of mitochondrial function [14,15,16,17,18,19,20]. The current state-of-the-art method for diagnosis of cellular rejection is performed by core needle biopsy and is analyzed under a light microscope using the grading scale of the Banff Schema, which is added to a final rejection index (RAI) [72]. While the Banff Schema indicates the presence and severity of acute cellular rejection, the RAI score has been criticized for showing an inadequate correlation with response to therapy or graft survival [73]. The correlation of graft performance and mitochondrial functional analyses of the OxPhos and ETC systems in the included experimental studies indicates the potential of high-resolution respirometry for quantitative assessment of allograft injury upon transplantation, providing a basis for diagnostic approaches and evaluation of improved preservation techniques for liver grafts [16,46]. In conclusion, our meta-analysis reveals that dynamic preservation of grafts with HMP and SNMP is superior to SCS with respect to protection of mitochondria. As presented, cold temperature appears to benefit mitochondrial metabolism; however, this protection disappears in prolonged storage (SCS ≥ 12 h). Consequently, a combination of dynamic preservation and a shorter storage time at low temperatures could potentially provide better mitochondrial protection in LT. In order to validate this, clinical studies on human patients are required and warranted in the future.

## 4. Materials and Methods

This study is reported in accordance with the PRISMA 2009 (Preferred Reporting Items in Systematic Reviews and Meta-Analysis) statement (Appendix A). The review protocol was registered with the National Institute for Health Research PROSPERO system under registration number 224134.

### 4.1. Search

The literature search was performed with a systematic search of the EMBASE, MEDLINE (via PubMed), Cochrane Controlled Register of Trials (CENTRAL) and Web of Science databases. No language limitation was applied. The date of the final literature search was 3 January 2021 (Appendix A).

### 4.2. Study Selection

When the duplicates were removed with reference manager software (EndNote X7), title and abstract screenings were performed on the remaining studies by one of the authors (T.H). After the selection, the full texts of potentially eligible records were obtained. Any questions in data extraction were settled by discussion with a second author.

### 4.3. Selection and Eligibility Criteria

Inclusion criteria specified any piece of literature comparing the preservation method, temperature and cellular changes (mitochondrial dysfunction and/or ATP level) that contribute to graft survival. Studies were excluded if they (i) did not include liver transplantations, (ii) liver enzymes, (iii) any information about mitochondrial changes, or (iii) did not report clinically relevant outcomes. The potential studies were identified using the previous search strategy. One author (T.H.) reviewed all the studies, and controversies were resolved by discussion with another author (P.H.). Full-text versions of potentially relevant studies were evaluated for inclusion using an eligibility pro forma screening document based on pre-specified criteria. The articles included the information obtained, the type of animal model, the outcome of the liver transplantation and the severity of mitochondrial damage.

### 4.4. Risk of Bias

We provide a Risk of Bias analysis for animal intervention studies (SYRCLE Risk of Bias analysis). This analysis is based on the Cochrane Risk of Bias analysis and has been adjusted for aspects of bias that play a specific role in animal intervention studies.

### 4.5. Outcomes

The ATP content of the graft was the primary outcome, as it indicates overall mitochondrial function. Secondary outcomes were the respiratory activity of mitochondrial complexes at different metabolic states of the mitochondria (OxPhos, Leak and RCR), cytochrome c and aminotransferase (AST) release. The outcomes of preservation methods (SCS, HMP, SNMP and MP) were compared with the control grafts of studies showing normal liver values without ischemic damage or with negligible ischemic damage. These grafts are relevant models for the clinical settings of living donor liver transplantation (LDLT). The grafts that underwent only WI without cold storage were considered as a separate group to investigate the effects of warm ischemic damage alone. This group thereby represents the WI damage of DCD grafts, which can be further superimposed by the cold ischemia damage during storage of the graft. 

### 4.6. Statistical Analysis

The network meta-analysis was conducted within the frequentist framework using the R package netmeta [74], which is a generalization of the pairwise meta-analysis that has been found to be equivalent to the most frequent approach used for network meta-analyses [75]. Random effect analysis was used for all comparisons. The input data for netmeta was the mean difference between the treatment arms and the corresponding standard errors (SE). Where the standard error of the difference was not explicitly given, it was calculated from the 95% confidence interval assuming a normal distribution.

## Figures and Tables

**Figure 1 ijms-22-02816-f001:**
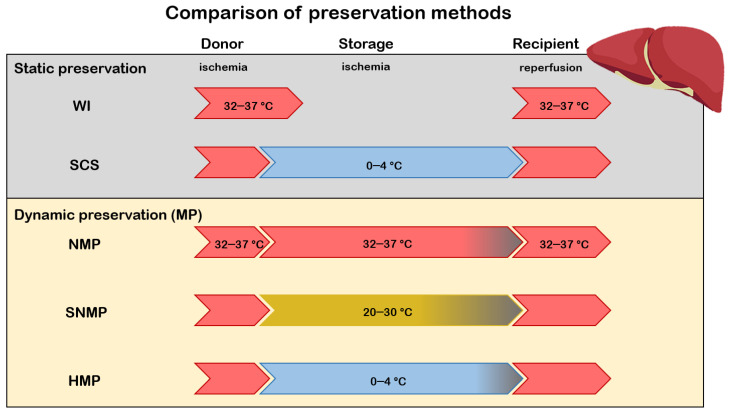
Comparison of different preservation methods. Upon transplantation, the graft undergoes consecutive stages of ischemia-reperfusion (IR). WI: liver grafts are exposed to warm ischemia to simulate donation after circulatory death (DCD); SCS: grafts after procurement from donors are exposed to cold ischemia during storage and organ transport. Dynamic preservation maintains organ perfusion using pulsatile machine perfusion (MP). MP can be divided into subgroups according to the temperature of the perfusate as normothermic machine perfusion (NMP), subnormothermic machine perfusion (SNMP) and hypothermic machine perfusion (HMP). The different colors of the arrows represent the temperature of the environment. Red indicates a 32–37 °C environment, blue demonstrates 0–4 °C, and yellow represents 20–30 °C.

**Figure 2 ijms-22-02816-f002:**
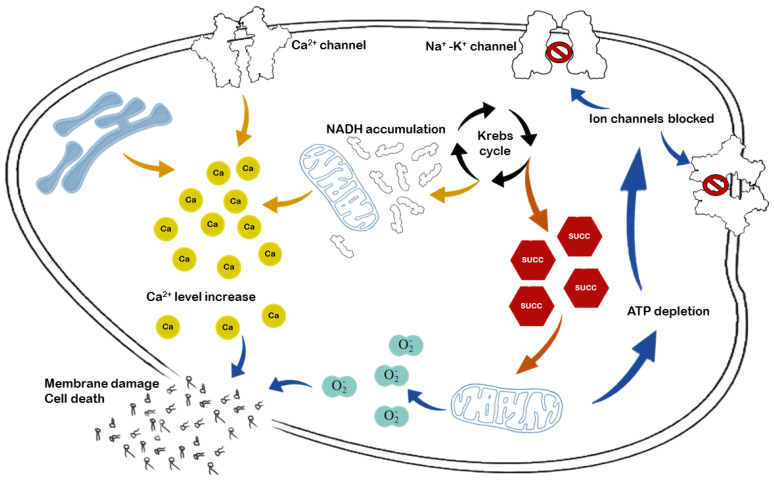
Intracellular mechanisms of ischemic injury are mediated mainly through ATP depletion. Oxygen deficiency results in impairment of oxidative phosphorylation and the production of reactive oxygen species (ROS), such as superoxide radicals (O_2_^−^). As an upstream mechanism, accumulation of succinate and NADH is responsible for increasing mitochondrial Ca^2+^ flux. As regards downstream changes, inhibition of ATP-dependent Ca^2+^ channels further increases intracellular Ca^2+^ levels and causes membrane damage and cell death in conjunction with ROS.

**Figure 3 ijms-22-02816-f003:**
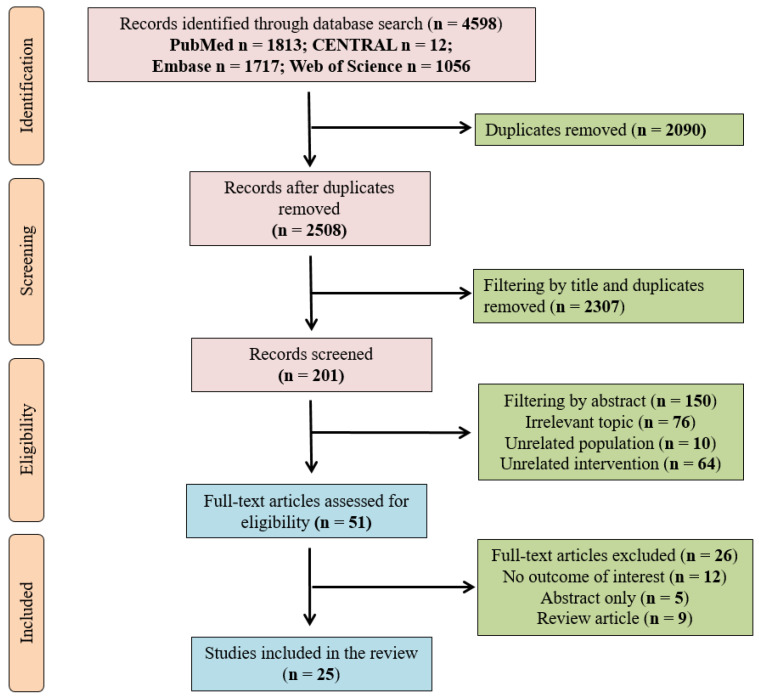
Flowchart.

**Figure 4 ijms-22-02816-f004:**
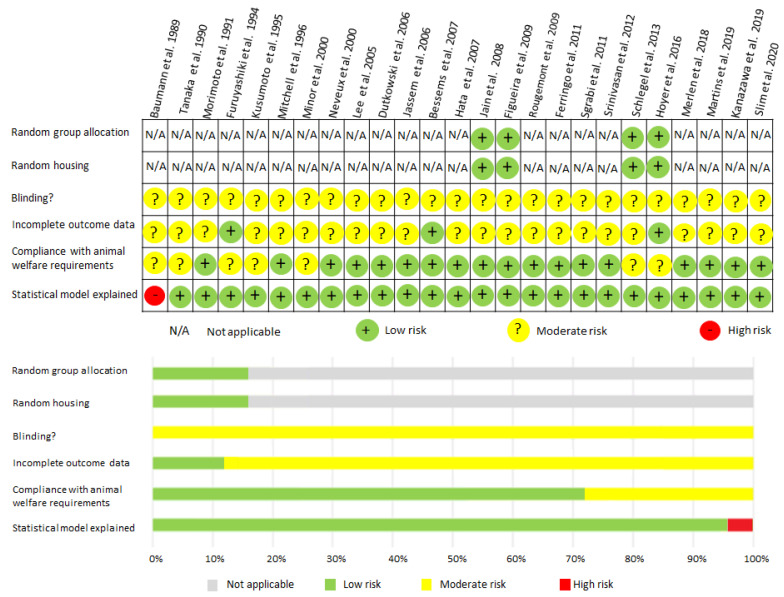
SYRCLE Risk of Bias analysis: Review of authors’ judgments about each risk of bias item for each study included.

**Figure 5 ijms-22-02816-f005:**
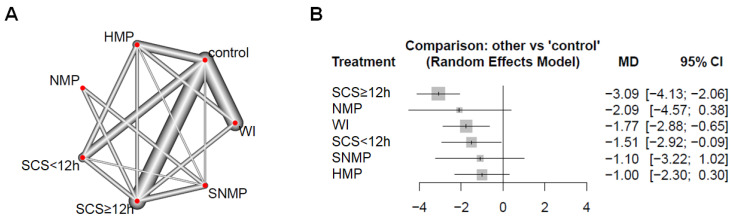
Comparison of tissue ATP content. (**A**) Networks of all preservation and control groups in included studies. The width of the lines is proportional to the number of direct comparisons. (**B**) Forest plot of mean differences (MD) compared to control group. For all comparisons, the random effect analysis was used. SCS ≥ 12 h: static cold storage for over 12 h; SCS < 12 h: static cold storage for less than 12 h; WI: warm ischemia; HMP: hypothermic machine perfusion; NMP: normothermic machine perfusion; SNMP: subnormothermic machine perfusion; CI: confidence interval.

**Figure 6 ijms-22-02816-f006:**
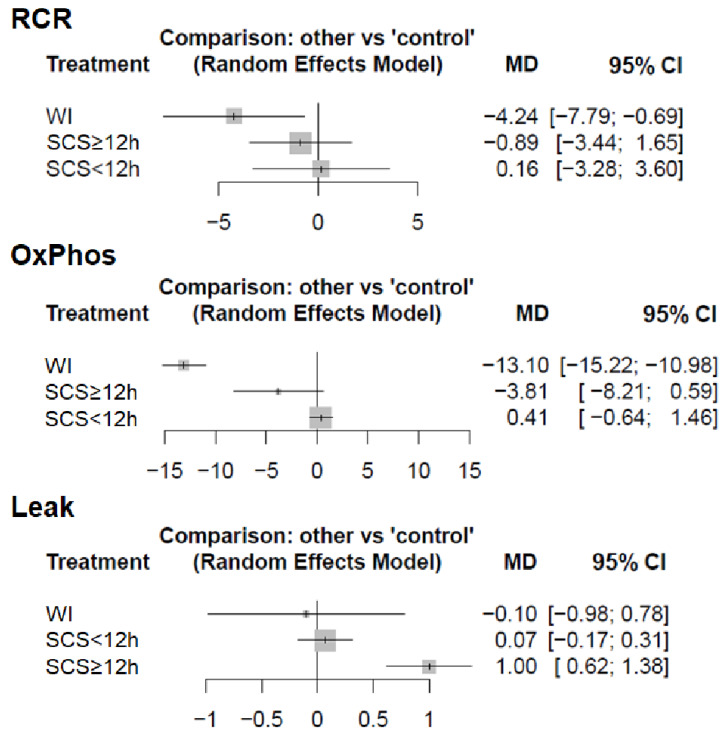
Comparison of complex I-related mitochondrial respiratory function following static cold storage. SCS ≥ 12 h: static cold storage for over 12 h; SCS < 12 h: static cold storage for less than 12 h; WI: warm ischemia; OxPhos: oxidative phosphorylation; Leak: leak respiration; CI: confidence interval.

**Figure 7 ijms-22-02816-f007:**
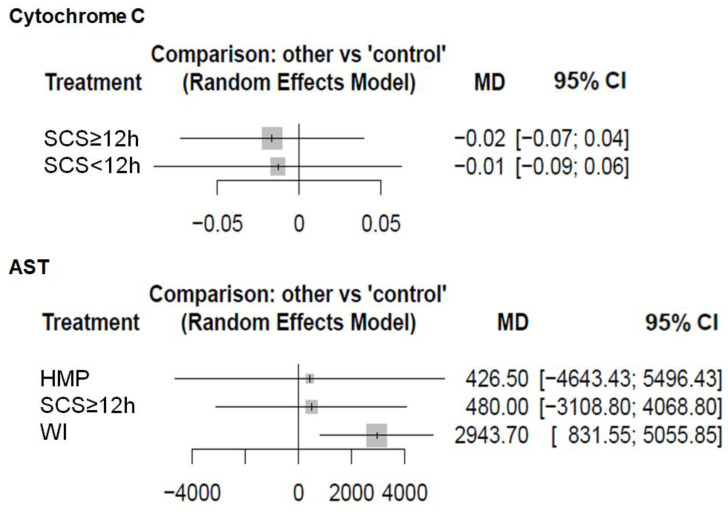
Comparison of cellular damage markers. Forest plot of mean differences (MD) compared to the control group. For all comparisons, the random effect analysis was used. AST: aspartate aminotransferase; SCS ≥ 12 h: static cold storage for over 12 h; SCS < 12 h: static cold storage for less than 12 h; WI: warm ischemia; HMP: hypothermic machine perfusion; CI: confidence interval.

**Figure 8 ijms-22-02816-f008:**
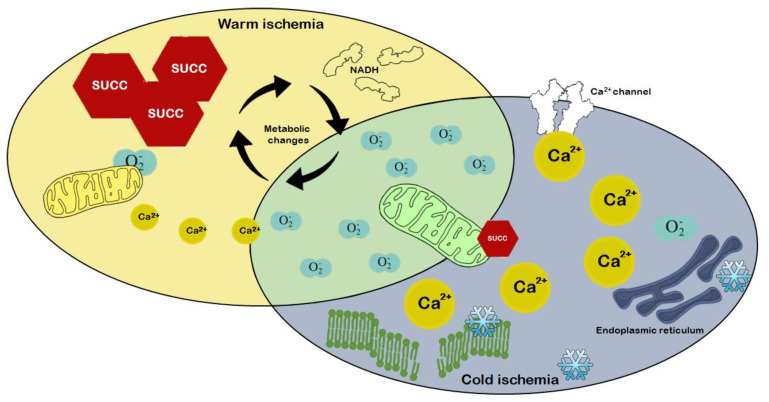
Intracellular changes in warm and cold ischemia. Warm ischemia is mainly characterized by an accumulation of succinate and NADH, while an increase in Ca^2+^ levels is more prominent in cold ischemia. Mitochondrial ATP depletion is more significant in warm ischemia, causing severe metabolic changes. Production of reactive oxygen species can be observed in both types of ischemia.

**Figure 9 ijms-22-02816-f009:**
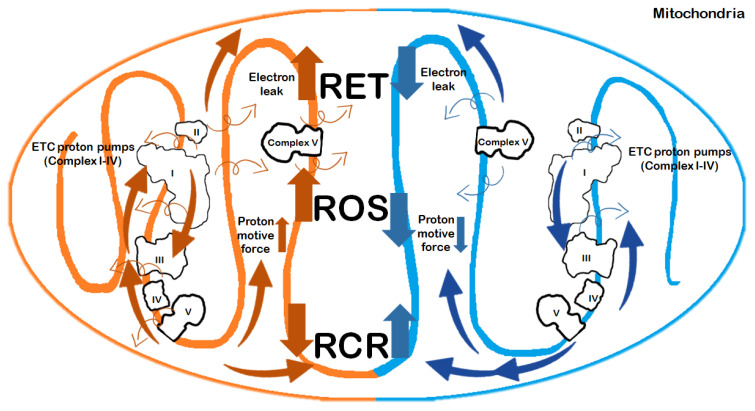
Changes in mitochondrial functions in warm and cold ischemia. Warm and cold ischemic changes are shown on the left and right side of the figure, respectively. Ischemia impairs complex V (ATP synthase). This indicates that electron transport is not coupled to ATP synthesis, and was decreased in response to prolonged preservation (SCS ≥ 12 h). The electron leak at mitochondrial complex I at the onset of reperfusion is the main source of oxidative stress [17]. The mechanism is initiated by the ischemic accumulation of the tricarboxylic acid (TCA) cycle metabolite succinate, which fuels the reverse electron transfer (RET) toward complex I, thereby leading to the production of reactive oxygen species (ROS) [14]. These findings raise the possibility that impairment of mitochondrial function may be an underestimated contributor to the compromised liver graft function after transplantation.

**Table 1 ijms-22-02816-t001:** Predictors of outcome in extended criteria donor (ECD) livers.

	Parameters	Variables
Donor characteristics	Age	>65 years
	ICU stay with ventilation	>7 days
	BMI	>30 kg/m^2^
	DCD	
Laboratory parameters	Serum Na^+^	>165 mmol/L
	Serum bilirubin	>3 mg/dL
	ALT/GPT	>105 U/L
	AST/GOT	>90 U/L
Histology	Steatosis of the liver	30–60% (macrosteatosis)
Preservation time	CIT	>10.5–14 h

Risk factors associated with higher postoperative complications, early allograft dysfunction (EAD) and primary graft non-function (PNF) refer to advanced donor age (over 65 years), steatosis of the liver (30–60% macrosteatosis), prolonged ICU stay with ventilation (over 7 days), body mass index (BMI) over 30 kg/m^2^, serum sodium (Na^+^) levels over 165 mmol/L, elevated alanine aminotransferase (ALT) and aspartate aminotransferase (AST) levels, increased serum bilirubin, donation after cardiac death (DCD) and long cold ischemic time (CIT) of procured graft.

**Table 2 ijms-22-02816-t002:** Characteristics of included studies (RCR: respiratory control ratios; TAN: total adenine nucleotid; GDH: glutamate dehydrogenase; MDA: malondialdehyde; LDH: lactate dehydrogenase)

Characteristic of Studies						Outcome Parameters		
Author	Year	Country	Species	Preservation Methods	Number of Cases	Primary	Secondary	
						ATP	Other Mitochondrial Parameters	Other liver Parameters
Baumann [29]	1989	Netherlands	rat	WI	ND	+	RCR, Oxphos, Leak	
Morimoto [30]	1991	Germany	rat	SCS	6	+	RCR, Oxphos, Leak	TAN,
Tanaka [31]	1990	Japan	rat	SCS,WI	5,5	+	RCR, Cytochrome C, Oxphos	TAN
Furuyashiki [32]	1994	Japan	rat	SCS	5	+		
Kusumoto [33]	1995	Germany	rat	SCS	6	+		TAN, LDH
Mitchell [34]	1996	Canada	rat	SCS	4	+		
Neveux [35]	2000	France	rat	SCS,HMP,SNMP	ND	+		AST, ALT, LDH
Minor [36]	2000	Germany	rat	SCS	5			ALT, LDH
Jassem [37]	2006	Italy	rat	SCS	6		Cytochrome C, Hydrogenperoxide	
Lee [38]	2005	South Korea	rat	WI	8	+	GDH, Hydrogenperoxide	MDA, TAN, AST, ALT
Dutkowski [39]	2006	Switzerland	rat	SCS, HMP	8,12		Cytochrome C, Caspase 3	MDA, LDH
Bessems [40]	2007	Netherlands	rat	SCS,HMP	7,7	+		AST, LDH
Hata [41]	2007	Germany	rat	SCS	7	+	GDH	AST, ALT, LDH
Jain [42]	2008	USA	rat	SCS,HMP	5,5	+		ALT, LDH
Figueira [43]	2009	Brazil	rat	WI	8		RCR, Oxphos	AST, ALT
Rougemont [44]	2009	Switzerland	pig	SCS,HMP	6,6	+		AST, ALT
Ferringo [45]	2011	Italy	rat	SCS,SNMP	5,5	+	GDH	AST, ALT, LDH
Sgrabi [46]	2011	Italy	rat	SCS	ND	+	Oxphos, Leak	
Srinivasan [47]	2012	Japan	rat	SCS	5		GDH	MDA, AST, ALT
Schegel [48]	2013	Switzerland	pig	SCS,HMP	8,8		Cytochrome C	AST,
Hoyer [10]	2016	Germany	pig	SCS,SNMP,NMP	5,6,5	+		TAN,
Merlen [49]	2018	Canada	rat	WI	5	+	Caspase 3	AST, ALT
Martins [16]	2019	Portugal	rat	SCS	4	+	RCR	
Kanazawa [50]	2019	Japan	pig	SCS,HMP,SNMP	5,5,5	+		AST, LDH
Slim [51]	2020	Tunisia	rat	SCS	6		Caspase 3, GDH	MDA, AST, ALT

## Data Availability

The data presented in this study are available on request from the corresponding author.

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
