# Peer review of "Mitochondrial Consequences of Organ Preservation Techniques during Liver Transplantation"

_ijms, 2021, doi:10.3390/ijms22062816_

Round 1

Reviewer 1 Report

In this report, the authors made an interesting meta-analysis about mitochondrial damage induced by different graft preservation techniques in liver transplantation surgery. After the analysis, they concluded that the combination of dynamic preservation and shorter storage at low temperature provide an optimal protection from mitochondrial damage for hepatocytes in liver transplantation. Overall, the report is interesting and offers a novel approach to be considered for optimal preservation of grafts during the surgery in order to assure an optimal conditions to avoid early graft failure or rejection. In my opinion, this report have enough scientific interest to be considered for its publication in this journal. My only minor concern about this report is that, due to the high numbers of abbreviations used in the text, an abbreviation sections should be included.

Author Response

Prof. Dr. Kurt A. Jellinger

Editor-in-Chief

Re: Ms. entitled “Mitochondrial Consequences of Organ Preservation Techniques During Liver Transplantation” by Tamara Horváth et al. (ijms-1110612)

Dear Professor Jellinger,

            Thank you very much for your letter of 24th February 2021 and for the constructive criticism of our paper. We thank you and the Reviewers for the valuable comments, which gave us the opportunity to clarify the novel aspects of our study and to improve the manuscript.

Reviewer 1.

1)         My only minor concern about this report is that, due to the high numbers of abbreviations used in the text, an abbreviation sections should be included.

            Thank you for addressing this issue. We added abbreviation section in to the main text (page 14, paragraph 8).

Reviewer 2 Report

In this manuscript by Horváth et al., the authors show the relevance of mitochondrial machinery alterations during cold ischemia and reperfusion.  The paper is clear,  well defined and structured. However, some aspects concerning to the  relevance of oncotic agent (PEG 35) on mitochondrial machinery and related  markers (ALDH2)  during  hypothermia static/dynamic (HOPE) strategies should be considered and commented.

Bibliography should be changed

Author Response

Prof. Dr. Kurt A. Jellinger

Editor-in-Chief

Re: Ms. entitled “Mitochondrial Consequences of Organ Preservation Techniques During Liver Transplantation” by Tamara Horváth et al. (ijms-1110612)

Dear Professor Jellinger,

            Thank you very much for your letter of 24th February 2021 and for the constructive criticism of our paper. We thank you and the Reviewers for the valuable comments, which gave us the opportunity to clarify the novel aspects of our study and to improve the manuscript.

Reviewer 2.

1)         However, some aspects concerning to the relevance of oncotic agent (PEG 35) on mitochondrial machinery and related markers (ALDH2) during  hypothermia static/dynamic (HOPE) strategies should be considered and commented.

Thank you for this comment. Accordingly, the following paragraph with the relevant reference (56) was added to the text (page 10, paragraph 3.1, line 245). The studies included in our meta-analysis did not provide sufficient information on ALDH2 changes, therefore we were not able to further consider the effects of oncotic agents in that case.